# Multiple Cofactor Engineering Strategies to Enhance Pyridoxine Production in *Escherichia coli*

**DOI:** 10.3390/microorganisms12050933

**Published:** 2024-05-03

**Authors:** Lijuan Wu, Jinlong Li, Yahui Zhang, Zhizhong Tian, Zhaoxia Jin, Linxia Liu, Dawei Zhang

**Affiliations:** 1School of Biological Engineering, Dalian Polytechnic University, Dalian 116034, China; wulj@tib.cas.cn (L.W.); zhangyahui@tib.cas.cn (Y.Z.); 2Tianjin Institute of Industrial Biotechnology, Chinese Academy of Sciences, Tianjin 300308, China; li_jl@tib.cas.cn (J.L.); tianzhzh@tib.cas.cn (Z.T.); zhang_dw@tib.cas.cn (D.Z.); 3National Center of Technology Innovation for Synthetic Biology, Tianjin 300308, China; 4Key Laboratory of Engineering Biology for Low-Carbon Manufacturing, Tianjin Institute of Industrial Biotechnology, Chinese Academy of Sciences, Tianjin 300308, China; 5University of Chinese Academy of Sciences, Beijing 100049, China

**Keywords:** pyridoxine, cofactor engineering, enzyme design, NAD^+^ regeneration

## Abstract

Pyridoxine, also known as vitamin B_6_, is an essential cofactor in numerous cellular processes. Its importance in various applications has led to a growing interest in optimizing its production through microbial biosynthesis. However, an imbalance in the net production of NADH disrupts intracellular cofactor levels, thereby limiting the efficient synthesis of pyridoxine. In our study, we focused on multiple cofactor engineering strategies, including the enzyme design involved in NAD^+^-dependent enzymes and NAD^+^ regeneration through the introduction of heterologous NADH oxidase (Nox) coupled with the reduction in NADH production during glycolysis. Finally, the engineered *E. coli* achieved a pyridoxine titer of 676 mg/L in a shake flask within 48 h by enhancing the driving force. Overall, the multiple cofactor engineering strategies utilized in this study serve as a reference for enhancing the efficient biosynthesis of other target products.

## 1. Introduction

Pyridoxine (PN), also known as vitamin B_6_ (C_8_H_11_NO_3_), is a water-soluble vitamin that was discovered during the study of rat acrodynia and was first synthesized in 1939 [1,2,3]. Vitamin B_6_ can be synthesized by bacteria, fungi, archaea, and plants, but animals can only obtain vitamin B_6_ from the diet due to the absence of the de novo biosynthetic pathways [4,5,6]. A deficiency in vitamin B_6_ can lead to various health problems, such as impaired immune function, dermatitis, anemia, and neurological disorders [7,8]. Therefore, it is crucial for animals, including humans, to consume an adequate amount of vitamin B_6_ through a balanced diet to meet their physiological needs [9].

With concerns about oil supply and environmental protection, there has been an increasing focus on developing green and sustainable pathways for chemical production [10]. In this regard, biosynthesis has emerged as a promising alternative. In our previous work, the bioproduction titer of PN reached 453 mg/L in a shake flask using mixed carbon resources produced by recombinant *E. coli* with two plasmids [11]. However, one of the challenges that arises during repeated microbial fermentations is the decrease in titer that occurs as the number of fermentation batches increases. This decline may be attributed to factors such as phenotypic variation, metabolic burden, or NADH (nicotinamide adenine dinucleotide, reduced form) imbalance [12,13]. When considering the biosynthesis of PN independent of cellular metabolic processes such as the glycolytic pathway, the production of one molecule of PN is accompanied by the generation of three molecules of NADH (Figure 1) [11,14]. Due to its high connectivity within the metabolic network, any change in the NADH/NAD^+^ ratio can lead to extensive metabolic changes [15,16]. Due to the excess of NADH in microbial cells, resulting in an imbalance in the NADH/NAD^+^ ratio, critical metabolic enzymes may be inhibited, leading to reductive stress and impairment in cofactor regeneration, which could potentially trigger strain degradation [17,18]. Therefore, researchers are committed to developing efficient cofactor supply systems and self-regulating cofactor systems [14,19].

One strategy employed is the mining of natural NAD(P)^+^-utilizing enzymes or protein engineering, which allows for obtaining NADP^+^-dependent dehydrogenases that exhibit catalytic properties similar to those of NAD^+^-dependent dehydrogenases [14]. We would prefer to obtain NAD(P)^+^-utilizing enzymes for the biosynthesis pathway of PN to reduce the consumption of NAD^+^. By replacing one enzyme with another, it becomes feasible to provide an ample supply of NADPH or NADH, consequently facilitating the biosynthesis of target products. In *E. coli*, for efficient glycolate production, the native *gapA* gene was replaced by introducing the NADP^+^-dependent glyceraldehyde 3-phosphate dehydrogenase encoded by *gapC* to ensure an adequate supply of NADPH [20]. Another approach involves introducing NADH oxidase (Nox) to reduce O_2_ to H_2_O_2_, or directly reducing O_2_ to H_2_O via a four-electron reduction, thereby catalyzing the oxidation of reduced NADH [21]. Due to the unique high activity of Nox from *Streptococcus pyogenes* (named SpNox) and the absence of by-products, this enzyme is coupled with a dehydrogenase to produce various L-rare sugars. It is a suitable option for cofactor NAD^+^ regeneration [22]. In *Bacillus subtilis* 168, an NAD^+^ regeneration system involving the expression of Nox was constructed to efficiently produce 91.8 g/L acetoin [23]. However, the overexpression of Nox can potentially disrupt the cellular demand for NADH or affect ATP production in the respiratory chain. An alternative key strategy to solely controlling the consumption of NADH is to substitute NADH production with NADPH production [22,24]. Therefore, to address the potential imbalance of cofactors in the biosynthesis pathway of PN, various strategies for regulating cofactor balance need to be attempted.

In this study, we combined multiple cofactor engineering strategies with enhanced driving metabolic force to efficiently produce PN. Through the introduction of the phosphoketolase (PKT) pathway, the first NAD^+^-dependent enzyme, Epd, enhanced the erythrose 4-phosphate (E4P) supply to increase the flow of precursors into PN biosynthesis. By employing a rational design strategy, further engineering of the last NAD^+^-dependent enzyme, PdxA, enhances the efficiency of PN production. Finally, by synergistically harnessing the oxidation of NADH and implementing a competitive enzyme to reduce NADH production, we successfully achieved an impressive titer of 676.6 mg/L, highlighting the effectiveness of our strategy in efficiently producing PN. Our results further advance the green manufacturing process of PN.

## 2. Materials and Methods

### 2.1. Strains, Media, and Growth Conditions

Information about the relevant strains in this study and the primers used are listed in Table 1 and Appendix A, respectively. The *E. coli* DH5α strain was used for plasmid construction and replication, and the engineered strains of the WL-series were derived from MG1655. During the construction of the strain, the culture was grown at 30 °C or 37 °C in Luria–Bertani medium (10 g/L bacto peptone, 5 g/L yeast extract, 10 g/L NaCl). Ampicillin (100 mg/L), kanamycin (50 mg/L), or chloramphenicol (34 mg/L) was added as required. Different concentrations of L-arabinose and isopropyl β-D-1-thiogalactoside (IPTG) were added for induction.

### 2.2. Fermentation Conditions

During fermentation, we took out the frozen strain stock solution from the −80 °C refrigerator, activated it on a solid petri dish with the corresponding resistance, and cultured it overnight. The next day, we selected a single colony and inoculated it into a test tube containing 5 mL of seed culture medium. The medium consisted of 10 g/L bacto peptone, 10 g/L glycerol, 5 g/L NaCl, and 5 g/L yeast extract. We adjusted the pH to 6.5 and added the appropriate amount of antibiotics. After incubating the test tubes for 12–16 h on a shaker at 37 °C, we transferred the culture from the test tubes to 24 deep-well plates or flasks containing FM1.4 medium. This medium consisted of 15 g/L glycerol, 1 g/L glucose, 5 g/L yeast extract, 5 g/L bacto peptone, 0.2 g/L MgSO_4_·7 H_2_O, 0.01 g/L MnSO_4_·5 H_2_O, 0.01 g/L FeSO_4_·7 H_2_O, 100 mM Na_2_HPO_4_·12 H_2_O (pH adjusted to 6.5), along with the necessary antibiotics. The initial inoculation amount was adjusted to an OD_600_ of 0.1. The parameters of a high-speed shaking incubator (Zhichu, Shanghai, China) were set to 37 °C, 800 rpm, and 80% humidity, and the 24-well plate was incubated under these conditions. For the PN production fermentation assay, the cultivation time was set to 48 h. All cultivations were performed in duplicate or triplicate.

### 2.3. Gene Overexpression, Deletion, and Site-Directed Mutagenesis

The genes (Appendix A) and oligo primers were synthesized by GENEWIZ Biotechnology Co., Ltd. (Suzhou, China). DNA purification, plasmid preparation, and gel recovery were performed using an Omega Bio-Tek kit. According to the instructions of the ClonExpress^®^ MultiS One Step Clone Kit (Vazyme Biotech Co., Ltd, Nanjing, China; catalog number C113-01), the recombinant plasmid was constructed using seamless cloning technology. Seamless Cloning [25], also known as In-Fusion Cloning, is a variation in traditional PCR product cloning. The method involves using 15–20 homologous bases at the end of both the vector and the primer. This ensures that the PCR product obtained from the two ends of the vector sequence has homology of 15–20 bases. The force of complementary base pairing is relied upon to form a loop without the need for enzyme linkage. This can be used for the direct transformation of the host bacterium. The linear plasmid can be repaired by the host bacterium’s enzyme system to fill the gap. To knock out and replace *E. coli* genes, the CRISPR-Cas9 system was utilized for traceless genome editing based on the pRed_Cas9_recA plasmid [26,27]. For the construction of the *pdxA* mutant plasmid, use pRSFDuet-1_pdxA_pdxJ1 as the template and introduce the mutation point using primers. To remove the complete template plasmid, the PCR product needs to be treated with DpnI at 37 °C for 90 min. The treated PCR products were transformed into competent cells, plated on LB plates, and incubated overnight. The mature single colonies were labeled, picked sequentially, and confirmed by PCR using validation primers. The primers consisted of a 20 bp sequence (Test-F) located approximately 150 bp upstream from the mutation site and a downstream sequence (Test-R) located approximately 1000 bp away from Test-F. We selected the appropriate size of DNA fragments using agarose gel electrophoresis. The selected fragments were then sent to GENEWIZ Biotechnology Co., Ltd. (Suzhou, China) for sequencing.

### 2.4. Determination of NAD^+^/NADH

Sample preparation: Approximately 10^6^ suspension cells were collected and washed with precooled PBS buffer. After centrifugation at 600× *g* for 5 min and removing all the supernatant, 200 μL of precooled NAD^+^/NADH extract was added to promote cell lysis. At 4 °C, a centrifugation at 12,000× *g* for 5 min was performed, and the supernatant was collected as the sample for testing. Determination of the total amount of NAD^+^ and NADH involved taking a sample of 20 μL and placing it in a 96-well plate. Next, we added 90 μL of ethanol dehydrogenase working solution, incubated it for 10 min, and then added 10 μL of chromogenic solution. We mixed evenly, incubated for 30 min at 37 °C, and measured the absorbance at 450 nm. Determination of NADH concentration: A 100 μL sample was transferred into a 1.5 mL centrifuge tube, heated at 60 °C for 30 min to decompose NAD^+^, and centrifuged at 10,000× *g* for 5 min; then, 20 μL of the supernatant was added to a 96-well plate. Subsequently, 90 μL ethanol dehydrogenase working solution and 10 μL chromogenic solution were added. The absorbance was measured after incubation for 37 min. Determination of NADH Standard Curve: The standard NADH concentrations of 0, 0.25, 0.5, 1, 2, 5, 8, and 10 μM were obtained through dilution. The absorbance was determined following the steps for NADH concentration determination. The standard curve was plotted with NADH concentration on the x-axis and absorbance on the y-axis. The concentrations of NAD^+^ and NADH were calculated based on the standard curve. A Beyotime NAD^+^/NADH detection kit (WST-8) was used.

### 2.5. GFP-Based Fluorescence Spectroscopy

The fluorescence of GFP was measured using a Neo2 (Bio-TeK) instrument. The excitation and emission wavelengths were set to 488 nm and 525 nm, respectively. Fluorescence measurements were repeated three times for the untreated cell suspensions. Strain WL09 was cultured overnight at 37 °C in LB broth supplemented with ampicillin (50 μg/mL). The next day, the strain was cultured in different tubes containing FM1.4 supplemented with 1 g/L fructose, 1 g/L glucose, 5 g/L glucose, 0.2 mM L-arabinose, or 0.8 mM L-arabinose, respectively. To investigate the repeatability of the GFP emission levels, fluorescence measurements were repeated three times using an independently cultured cell suspension.

### 2.6. Analytical Methods

The Thermo Fisher High-Performance Liquid Chromatograph (HPLC) (Thermo Fisher Scientific, Waltham, MA, USA) and UltiMate 3000 system with an FLD-3400 detector and ODS column (Cosmosil AR-II; 250 × 4.6 mm, particle size 5 μm; Nacale Teske) were used to quantify the PN concentration. PN was separated by a gradient program. Mobile phase A consisted of 33 mM phosphoric acid and 8 mM 1-octane sulfonic acid. The pH was adjusted to 2.5 with KOH. Mobile phase B consisted of 80% acetonitrile (*v*/*v*). The total flow rate was set to 0.8 mL/min according to the literature method. The chromatographic supernatant was filtered through a 0.22 μm pore size filter membrane. The cell density was measured by optical density at 600 nm (OD_600_). We used the biosensor analyzer (M-100, Shenzhen Sieman Technology Co., Ltd., Shenzhen, China) to detect glycerol and glucose in the fermentation medium.

### 2.7. Enzyme Engineering Optimization

The complex structure of PdxA and the mutant complex bound to HTP and NAD^+^ was simulated using the PDB entries 1PS6 and 6XMY as reference structures [28]. AMBER20 [29] was used for molecular dynamics (MD) simulations. From the MD simulation trajectory, a representative conformation was selected as a starting point for subsequent enzyme design based on transition state analog (TSA) [30]. The transition state analog (TSA) was constructed by covalently linking HTP and NAD^+^ guided by catalytic mechanisms [31]. Using the Rosetta enzyme design application, we redesigned approximately 6 Å in TSA. The experimental data guided two rounds of study. In each round, specific selected residues underwent saturation mutation, and the mutation with the most favorable binding energy was chosen for experimental verification. The comprehensive method used is listed in the Appendix A [29,32,33,34,35,36,37,38,39].

### 2.8. Validation of Plasmid Stability

The strains were cultured in a fermentation medium, and samples were taken at 0 h, 24 h, and 48 h, respectively. After sampling, the fermentation samples were plated onto LB plates and incubated at 37 °C for 16–20 h to allow colony formation. Subsequently, 100 randomly selected single colony points were inoculated onto corresponding antibiotic resistance plates for each strain. These microorganisms were then cultured at 37 °C until colonies grew. The stability of the plasmid (%) was calculated by determining the ratio of the number of colonies on the growth plate to the total number of spots.

## 3. Results and Discussion

### 3.1. Precursor Supply Enhancement by the Introduction of Phosphoketolase

Phosphoketolases catalyze the redox-independent cleavage of certain sugar phosphates, such as D-xylulose 5-phosphate (Xu5P) and D-fructose 6-phosphate (F6P), resulting in the formation of acetyl phosphate and the corresponding aldose phosphate, namely D-glyceraldehyde 3-phosphate (G3P) and E4P [40]. This enzyme has been extensively used in metabolic engineering to increase levels of acetyl coenzyme A, poly-(3-hydroxybutyrate), and aromatic chemicals [41,42]. To increase the precursor pool and enhance Epd catalysis since its *K*m for E4P is 0.737 mM, the heterologous *xfp* gene (encoding xylulose 5-phosphate/fructose 6-phosphate phosphoketolase) from *Bifidobacterium longum* was introduced into the LL006 strain, which was derived from a previous study [11]. This strain is based on *E. coli* MG1655, with the *pdxST* from *B. subtilis* and *pdxP* from *E. meliloti* inserted into the *pdxH* and *pta* locus, respectively. To obtain a natural supply of PLP, the strain integrated the *pdxST* gene, which encodes the enzyme responsible for direct PLP synthesis in *B. subtilis*, and the *pdxP* operon, which is composed of *Ensifer meliloti* in wild-type *E. coli* (MG1655) using the CRISPR Cas9 system [11].

Overexpressing *xfp* in LL006 (named WL01) resulted in a slight decrease in cell growth but led to an increase in PN production (Figure 2). We further tested the effect on the PN titer by expressing two plasmids harboring entire enzymes of the vitamin B_6_ pathway [p15ASI-Ptac-*epd* (Gni)-*pdxB* (Eco)-*dxs* (Eme)-P_J231119_-*serC* (Eco) and pRSFDuet-1_P3-*pdxA2* (H136N)-*pdxJ1* (E104T/I218L/G194C)] [11]. The results showed that the integrated *xfp* significantly enhanced PN production by 24% after 48 h of cultivation (Figure 2, Appendix A). The results indicated that the introduction of the PKT pathway was effective in enhancing PN production. This pathway has the potential to redirect flux toward precursors, primarily E4P or G3P, leading to an increased titer of the desired product, as previously reported [43].

### 3.2. Improve the Catalytic Efficiency of the Rate-Limiting Enzyme pdxA through Rational Design

PdxA is a rate-limiting enzyme in the biosynthesis pathway of vitamin B_6_, and previous studies have demonstrated that the overexpression of this enzyme can enhance production [11]. To further enhance the metabolic driving force for PN biosynthesis and optimize the binding capacity between 4HTP and NAD^+^ in an approximate catalytic state, a transition state analog (TSA) model was employed, where 4HTP and NAD^+^ were covalently bound. As described in the computational methods section, we conducted two rounds of design on 71 residues (Appendix A) within a 6 Å range round of TSA. In the first round, we conducted saturation mutagenesis simulations, resulting in the design of 216 mutants with mutations in 62 residues. Experimental validation was conducted on mutants at each residue position with the most negative binding energy and binding score lower than the wild type (WT). A total of 12 mutant strains with increased potency in PN were obtained after screening 24 deep-well plates. Among them, the F140I mutant and the T165C mutant showed the most significant enhancement with titer increases of 72% and 40%, respectively (Appendix A). Using these promising mutants (which showed enhanced production in the first round of experiments) as templates, a second round of iterative saturation mutagenesis was performed on the remaining residues within a 6 Å radius of the TSA. Consistent with the selection criteria used in the first round, the combinations with the most negative binding scores, below that of the WT, were chosen for experimental validation from each group. In total, 104 combination mutants were selected. We successfully constructed 83 mutants, out of which 40 mutants exhibited a preliminary increase in titer of 10% or more during the screening using 24-deep well plates (Appendix A). Subsequently, we compared the fermentation performance of eight mutants that exhibited significant improvements in titer through single-point and double-point mutations in comparison with the WT and H136N mutant as presented in the previous study [11]. The comparison was conducted in 250 mL shake flasks containing 30 mL cultures. However, even the highest-producing double-point mutant did not surpass the titer of the F140I mutant (Figure 3A). The F140I mutation resulted in a 26% increase in titer, surpassing the productivity achieved by the previously reported H136N mutant (Figure 3A, Appendix A).

To investigate the potential impact of the F140I mutation on *pdxA*, we conducted additional molecular dynamics simulations to analyze the binding affinities of 4HTP and NAD^+^ in the pre-catalytic state. Specifically, we focused on parameters where Dis_1 < 3.5 Å (Dis_1: 4HTP_HB-NAD_C42) and Ang_1 > 150° (4HTP_CB-4HTP_HB-NAD_C42). Remarkably, the binding affinity has significantly improved (F140I = −208.85 kcal/mol, WT = −183.84 kcal/mol). Further trajectory analysis indicates that 4HTP exhibits a more stable binding pattern, whereas NAD^+^ experiences progressively larger fluctuations throughout the dynamic simulation (Figure 3B). Following the introduction of the mutation (F140I), 4HTP displayed reduced oscillations (RMSD from 1.8 Å–3.6 Å to 1.03 Å–2.54 Å), and NAD^+^ tended to stabilize within a specific range of fluctuations. Analysis of the representative conformations revealed that the WT, NAD^+^ primarily engages in interactions with the ribose and amide groups of nicotinamide. Conversely, the F140I mutant displayed preferential binding to the pyrophosphate and adenine regions. Furthermore, the F140I mutation facilitated the inclusion of an additional hydroxyl group in the ion coordination, resulting in tighter binding. Additionally, the angle between 4HTP_HB-NAD_C42 and the catalytic atom of NAD_C42 increased from 107.0° in the WT to 166.44° in the F140I mutant (Figure 3C,D) with favorable impacts on catalytic efficiency. The refined angles and differential binding affinities are likely to contribute to the enhanced catalytic activity of the F140I mutant, thereby increasing the metabolic flux of PdxA and consequently boosting PN production.

Moreover, the F140I mutation, while improving the catalytic angle, resulted in an adverse increase in the catalytic distance (from 3.5 to 4.5 Å; see Figure 3D). This indicates that in the dynamics of the process, NAD^+^ is unlikely to form a pre-catalytic state with 4HTP in most cases. Due to the mobility of both the protein and the ligand, this situation could potentially be overcome through fluctuations of the protein and substrate, thereby reaching a pre-catalytic state and completing catalysis. However, the probability of this state occurring in both the wild-type and mutant MD trajectories is less than 10% (WT: 6.9%, F140I: 3.0%). This suggests that although fluctuations can help NAD^+^ overcome distance or angular challenges to form a pre-catalytic state, excessively flexible structures still hinder the catalytic stability and efficiency of the protein [44]. Therefore, increasing the supply of NAD^+^ in vivo to raise its relative concentration might allow more NAD^+^ to enter the active site. This, in turn, could increase the probability of forming a pre-catalytic state and ultimately enhance the overall catalytic efficiency of pdxA. Consequently, the effective binding and sustained provision of NAD^+^ potentially emerge as pivotal constraints for the further catalysis of PdxA. Hence, given these factors, we pursued NAD^+^ cofactor optimization to ensure an ample supply of PdxA.

### 3.3. Cofactor Engineering by Leaky Expression of SpNox for PN Production

We examined the intracellular levels of NADH and NAD^+^ in our study. Interestingly, a significant decrease in both the concentrations of the NAD^+^ and NADH at 48 h was observed in the high-production strain (WL03) compared to those in the WT (Figure 4A). These findings indicate a potential shift in the redox balance within the engineered strain, suggesting alterations in cellular metabolism and potential implications for PN production. Additionally, we observed a decrease in the overall concentration of NADH under the tested conditions (Figure 4A). These unexpected findings suggest that there may be additional factors influencing the NADH levels in the system. One possible reason for the observed reduction in the overall concentration of NADH could be that the NADH cofactor produced was tightly bound to PdxB or utilized in various cellular processes, such as energy production, biosynthesis, and redox balance [3]. This utilization resulted in a decrease in the available pool of NADH. To ensure optimal carbon flux toward target metabolites and maintain redox stability, it is desirable to employ strategies that can effectively regulate NADH levels in microbial systems. In our study, we addressed this issue by introducing Nox, which is an enzyme that facilitates the regeneration of NAD^+^ by oxidizing NADH.

Nox can effectively regenerate NAD^+^ oxidized cofactors because of its high activity with NADH [22]. To achieve redox homeostasis during the production of PN, we introduced SpNox into strain WL03 to obtain strain WL04. In WL04, the *pdxA2* gene was replaced with the newly screened *pdxA* mutation (F140I). A common method for controlling gene expression is to clone the gene under an inducible promoter. Therefore, a vector containing the tunable L-arabinose operon promoter *P_BAD_* was developed to fine-tune the expression of SpNox in WL04. Compared with WL03, the concentration of NADH decreased in WL04, while the concentration of NAD^+^ increased significantly. This study demonstrated the expression of SpNox in the high-titer strain WL04 (Figure 4A). To fine-tune the expression of SpNox, different concentrations of L-arabinose were used on *E. coli* after 18 h of cultivation. The fermentation medium used was FM1.4, which was supplemented with 15 g/L glycerol and a small amount of 1 g/L glucose as a carbon source. The timing of induction is determined based on the hypothesis that the early expression of SpNox may lead to a reduced level of NADH, which is crucial for cell growth. By introducing L-arabinose at a later stage, after the depletion of glucose in the medium, it is possible to circumvent the adverse effects of carbon catabolite repression (CCR). The results showed that the leaky expression of SpNox was adequate for a significant increase in the PN titer, which was comparable to that of the inducer (Figure 4B). These findings suggest a relationship between basal expression and induction capacity.

We measured the expression level of SpNox by comparing it to the targeted level of GFP replaced using the same strain (Figure 4C). Leaky expression from pBAD was not prevented by the addition of 1 g/L glucose to the fermentation media. The fluorescence values measured without the addition of the L-arabinose solution were used as the background fluorescence values. Compared with the group that added 0.2 mM L-arabinose, the leaky expression of the control group was almost 64.3% in 16 h. As shown in Figure 4C, the relative fluorescence exhibited a dose-dependent response to the increasing concentration of L-arabinose and a decreasing trend over the cultivation period. This was thought to be due to the catabolism of the L-arabinose used for induction. On the other hand, the background fluorescence represented a basal or leaky expression of the pBAD-controlled GFP expression system. It was previously reported that the expression of pBAD was significantly suppressed by fructose or glucose. In our experiment, we observed that the expression of GFP fluorescence/OD_600_ was significantly suppressed by 5 g/L glucose but not by 1 g/L glucose or 1 g/L fructose (Figure 4C). Thus, we used various concentrations of glucose in the fermentation medium. The results showed that WL04 produced 479.39 mg/L PN using 5 g/L glucose (Figure 4B, Appendix A). It indicates that a lower expression of SpNox is more beneficial for PN production.

The leaky expression of SpNox in a high-production strain was found to be beneficial for PN production. This benefit arises from the fact that the expression of SpNox needs to be restricted to a low level in order to maintain sufficient levels of NADH within the cell. This is necessary to ensure that enough NADH enters the respiratory chain for ATP production. Therefore, maintaining the expression of Spnox at a low level is necessary. In addition, the presence of three plasmids in the bacterial strain may result in strain instability, as confirmed by inconsistent production levels. To study the genetic stability of the PN-producing *E. coli* populations, the engineered strain WL04 harboring three plasmids was tested through metabolic evolution, which was achieved by sequentially subculturing the strain under constant antibiotic selection. The selection process started by using FM1.4 medium with serial transfers at 12 h intervals to achieve populations exceeding 100 cell generations (Appendix A). A significant decrease in PN titer was observed during the 44th to 100th generations (Figure 4D). Approximately 22% of the cells experienced plasmid loss, which was determined by comparing the percentage of colonies observed on LB plates with antibiotics to those on non-antibiotic plates at a specific time point (Appendix A). The growth rate of the mutant strains decreased with each successive generation (Figure 4E). Thus, we integrated the SpNox gene into the genome to enhance its genetic stability, which is advantageous for long-term fermentation processes.

To further decrease the SpNox expression and strain instability, the multi-copy nature of the pBAD plasmid was avoided through genome fusion combined with further promoter optimization. The gene encoding SpNox was inserted into the *ykgA* locus. To avoid the use of inducers, we utilized two constitutive promoters to initiate transcription. One was the constitutive Biobrick J23118 promoter with low expression (http://parts.igem.org/Part:BBa_J23118, accessed on 19 October 2023), and the other was synthesized using the Promoter Calculator online tool (https://www.denovodna.com, accessed on 19 October 2023). Considering the difference in copy number between the plasmid and genome, the targeted transcription rate at a single transcriptional start site (forward direction only) of the synthesized promoter (named Pro) was 3- to 5-fold higher than that of the J23118 and *P_BAD_* promoters, respectively. Next, we explored the effect of the difference in the PN titer between the SpNox leaky expression and constitutive expression in the genome. To evaluate this possibility, the two plasmids containing all enzymes of the vitamin B_6_ pathway were reintroduced into the strains WL06 and WL07 to generate WL08 and WL09, which were then cultured for PN testing. The results revealed that the redesigned promoter had a more advantageous effect on PN production compared to the other promoters (Figure 4F). Furthermore, compared with WL04, the concentration of NAD^+^ in WL09 was significantly increased, while the concentration of NADH decreased. This result aligns with our expectation that more NADH is converted into NAD^+^ (Figure 4A). These findings indicate that enhancing the transcription intensity of the promoter within the genome could help offset the difference in titer resulting from a decrease in the genome’s copy number compared to that of the plasmid.

### 3.4. Synergistic Effects of the Introduction of NADPH-Dependent GapN or GapC

GapA (G3P dehydrogenase A) is one of the main enzymes responsible for NADH production in bacterial cells [45]. GapA is involved in the sixth step of glycolysis, converting G3P to 1,3-diphosphoglycerate (1,3-BPG) while generating NADH in the process [46]. This enzyme is widely present in many bacteria, including *E. coli* and other common strains [47]. To decrease the intracellular NADH pool in the high-production PN strain, we introduced the NADP^+^-dependent G3P dehydrogenase GapC from *Clostridium acetobutylicum*. This enzyme helps generate NADPH instead of NADH during the oxidation of G3P in glycolysis (refer to Figure 1) [20,48]. Additionally, the non-phosphorylating NADP^+^-dependent glyceraldehyde 3-phosphate dehydrogenase (GapN) from *Streptococcus mutans* serves as an alternative source of NADPH production by catalyzing the irreversible oxidation of G3P to 3-phosphoglycerate (Figure 1) [49,50]. These two enzymes were inserted into the *pflB* locus using the CRISPR-Cas9 system, respectively. We obtained two strains, WL156 (WL07, Δ*pflB*::*gapN*) and WL157 (WL07, Δ*pflB*::*gapC*). After introducing two plasmids containing the entire vitamin B_6_ biosynthetic pathway, the newly obtained strains WL158 and WL159 achieved PN production titers of 652.0 mg/L and 676.6 mg/L, respectively (Figure 5), after a 48 h fermentation period. It was demonstrated that the expression of NADP^+^-dependent GapN/C, combined with NADH oxidation, had a synergistic effect on the production of PN. The titer was significantly higher than before (Figure 5). It is worth noting that compared with WL09, the concentration of NAD^+^ in WL158 and WL159 was further increased, especially in WL159. This may be due to the insertion of *gapN* or *gapC*, which reduced the utilization of NAD^+^ by GapA, thus increasing the concentration of NAD^+^. This requires us to further explore the relationship between the level of NADP^+^ and NAD^+^.

The insertion of GapN or GapC competes with GapA for G3P, thereby reducing the metabolic flux to GapA and subsequently decreasing intracellular NADH production [49,51]. This reduction in NADH levels helps to alleviate NADH inhibition and restore a more balanced redox state within the cell. Unlike GapC, which only affects NADH production, the insertion of GapN disrupts ATP production. However, it is important to note that a certain amount of ATP is produced by the oxidation of NADH through the NADH oxidative respiratory chain. Therefore, the decrease in ATP production caused by GapN insertion may synergize with the increase in NADH consumption, leading to an imbalance in the regulation of the NADH pool in cells. As a result, the titer of the GapN-inserted mutant was slightly lower than that of the GapC-inserted mutant. Furthermore, in addition to its role in maintaining redox balance, a decrease in NADH levels and subsequent increase in NADPH production can have additional implications for various cellular activities. For example, α-KG has good solubility, and it can be quickly converted into L-glutamate by the NADPH-dependent enzyme L-glutamate dehydrogenase (R00248). This suggests that NADPH may be involved in this transformation, leading to an increase in the production of PN. The specific mechanisms by which NADPH may impact the transformation of α-KG to glutamine and potentially influence PN production require further investigation and understanding. NADPH may be involved in this transformation, leading to an increase in PN production. The specific mechanism requires further study and understanding.

Theoretically, following the conservation of carbon atoms, it can be approximated that in the absence of PN generation, one molecule of glycerol or glucose in the glycolysis pathway yields two molecules of NADH (Figure 1). Under the conditions of producing one molecule of PN, if glycerol is utilized as the primary carbon source, a scenario similar to our actual situation arises. By working backward from the biosynthetic pathway, a rough estimation suggests that the production of one molecule of PN generates approximately seven molecules of NADH simultaneously from three molecules of glycerol (Figure 1). Consequently, it can be theoretically concluded that the PN pathway generates more NADH compared to the glycolytic pathway. However, experimental testing revealed contrasting results regarding the consumption of glucose and glycerol during the fermentation process of strain WL09 [WL01, *ΔykgA*::Pro-SpNox, p15ASI-Ptac-*epd* (Gni)-*pdxB* (Eco)-*dxs* (Eme)-P_J231119_-*serC* (Eco), pRSFDuet-1_P3-*pdxA*3 (F140I)-*pdxJ*1)]. The fermentation medium contained 1 g/L glucose (~5.6 mM) and 15 g/L glycerol (~162.9 mM). At 48 h, the PN titer was 582.4 mg/L (Figure 4F), equivalent to 3.44 mM PN, while both glycerol and glucose were completely consumed as detected by biosensor analyzer. Therefore, it can be roughly estimated that PN synthesis produced 17.2 mM NADH, while glycolytic NADH can reach 337 mM under conditions where no NADH production occurs. The large production of NADH results in the decrease in intracellular NAD^+^ and imbalance of NADH (Figure 4A). In this article, this issue was addressed by cofactor engineering for the regeneration and balance of NAD^+^ and NADH. Experimental results confirmed the effectiveness of our design, with the titer before and after cofactor engineering increasing from the initial 367.4 mg/L (WL10) to 676.6 mg/L (WL159, Appendix A), representing an increase of 84%.

In addition, the assay of strains stability between WL03, WL04, WL09, and WL159 was conducted in the fermentation process. As depicted in Figure 5B, we observed that the WL04 strain, carrying three plasmids, exhibited a plasmid stability of 71% at 48 h, which is lower than that of the other strains. The newly constructed strains containing two plasmids exhibited better stability (>80%) compared to those containing three plasmids. This also underscores the significance of genome fusion in reducing the number of plasmids for strain stability. The strains resulting from the cofactor engineering serve as a foundational strain for the subsequent development of high-yield and stable strains.

## 4. Conclusions

In this study, a comprehensive cofactor engineering approach was proposed, targeting both the cofactor itself and the cofactor-dependent enzymes involved in the synthesis of PN. Several strategies have been implemented to enhance PN production. First, the introduction of the PKT pathway aimed to increase the pool of E4P, which enhanced the production capacity of PN. Second, the mutant design of the PdxA enzyme improved the binding affinity of 4HTP and NAD^+^ in the pre-catalytic state. Further investigations revealed the significance of NAD^+^ binding with PdxA, suggesting that enhancing NAD^+^ availability could be beneficial for the PN pathway. Furthermore, the heterologous expression of SpNox, an NAD^+^ regeneration enzyme, was introduced to further enhance NAD^+^ availability. A synergistic effect was achieved by replacing the NADH-dependent GapA enzyme with NADPH-dependent GapN or GapC, thereby optimizing the redox balance and promoting PN production. This study improved the PN growth performance of the strain through cofactor engineering. However, the optimal NADH/NAD^+^ ratio for PN production by this strain remains unclear. The conditions explored in this study alleviated the imbalance of NADH/NAD^+^, and new methods are still needed to determine the most suitable NADH/NAD^+^ ratio for producing PN in the later stage. Overall, this study successfully implemented an effective strategy that combined cofactor engineering on the cofactor itself and cofactor-dependent enzymes. This approach led to a significant increase in PN production, reaching up to 676.6 mg/L. These findings provide valuable insights for the development of efficient strategies for PN synthesis.

## Figures and Tables

**Figure 1 microorganisms-12-00933-f001:**
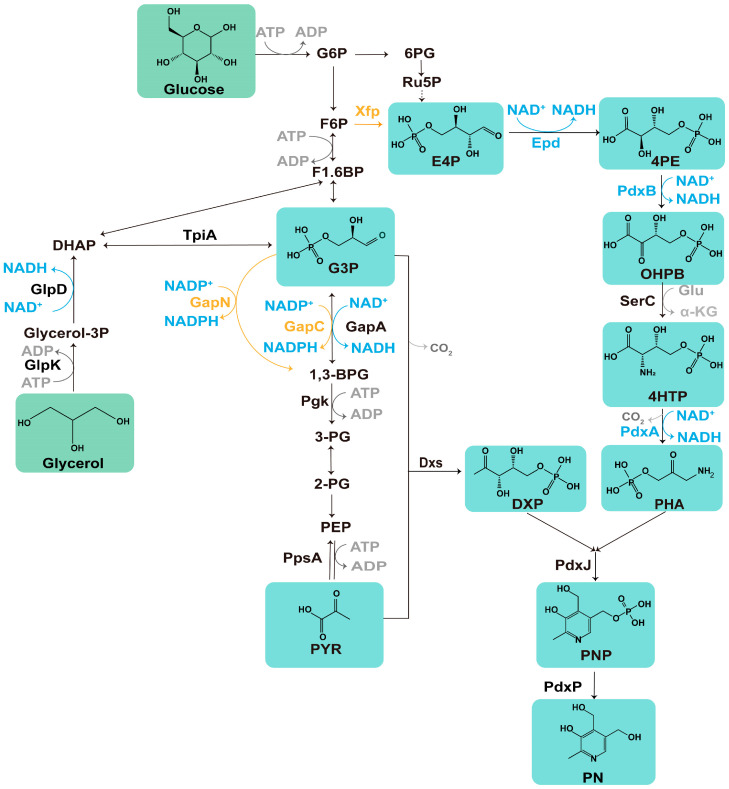
Scheme of the biosynthesis pathway of PN in *E. coli*: glycerol and glucose serve as carbon sources for the production of PN. The orange line represents the heterologous gene introduced in this study. The blue line, along with the accompanying words, illustrates the cofactor-related pathways and enzymes involved. The green box displays the chemical structure of the compounds involved in PN synthesis. Epd, D-erythrose-4-phosphate dehydrogenase; PdxB, erythronate-4-phosphate dehydrogenase; PdxA, 4-hydroxythreonine-4-phosphate dehydrogenase; GapA, glyceraldehyde-3-phosphate dehydrogenase A; GapC, NADP^+^-dependent G3P dehydrogenase; GapN, NADP-dependent glyceraldehyde 3-phosphate dehydrogenase; SerC, 3-phosphoserine aminotransferase; Xfp, xylulose 5-phosphate/fructose 6-phosphate phosphoketolase; PdxJ, PNP synthase; PdxP, PNP phosphatase; GlpD, glycerol-3-phosphate dehydrogenase; GlpK, glycerol kinase; TpiA, triosephosphate isomerase; Pgk, phosphoglycerate kinase; PpsA, phosphoenolpyruvate synthase.

**Figure 2 microorganisms-12-00933-f002:**
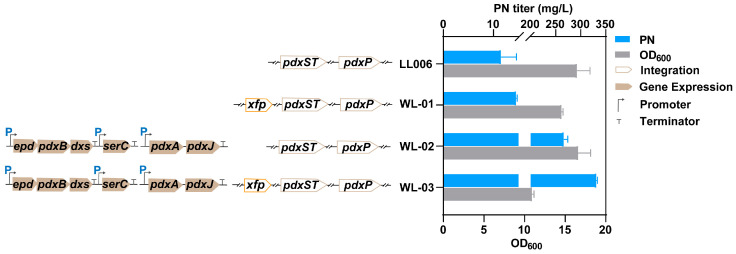
Scheme of the biosynthesis pathway of PN in *E. coli* and the PN titer and cell growth of the introduced *xfp* strains: PN titer and cell growth of the introduced *xfp* strains. The data are means ± SD of three independent biological replicates (*n* = 3).

**Figure 3 microorganisms-12-00933-f003:**
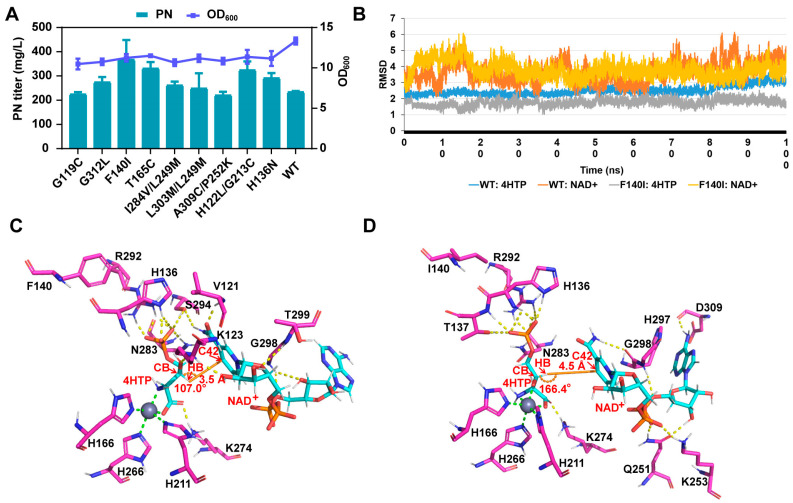
Enzyme design of PdxA. (**A**): The PN titer and cell growth of PdxA mutants with single and double mutations. The data are means ± SD of three independent biological replicates (*n* = 3). (**B**): The root-mean-square deviation (RMSD) variations were assessed for the substrate 4HTP and the coenzyme NAD^+^ in both the WT and F140I mutant strains throughout the molecular dynamics (MD) process (100 ns production trajectory) in comparison to the initial structures. (**C**,**D**): The intermolecular interaction networks between the substrates (4HTP and NAD^+^) and proteins were investigated in the representative conformations of the WT and F140I mutant strains. Hydrogen bond interactions are denoted by yellow dashed lines, metal coordination bonds by green dashed lines, and distance and angle relationships by orange lines. Protein residues are highlighted in magenta, while ligands are represented in cyan.

**Figure 4 microorganisms-12-00933-f004:**
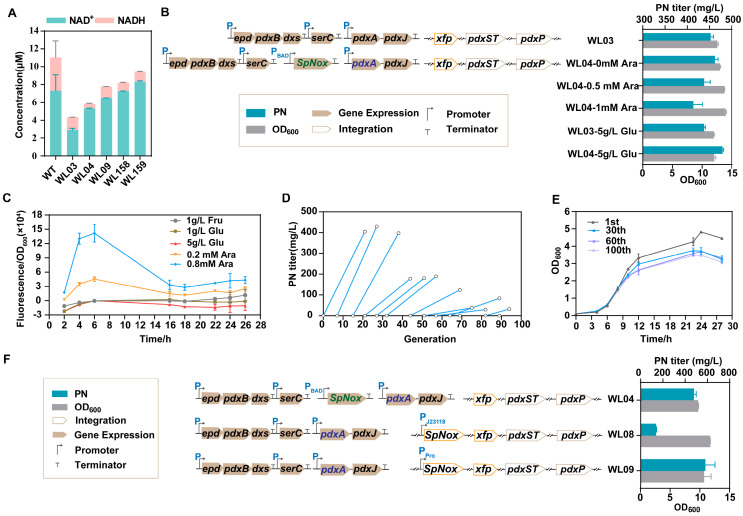
Cofactor engineering through leaky expression of SpNox for PN production. (**A**): The intracellular concentrations of NAD^+^ and NADH for all the main strains. (**B**): PN titer and cell growth (OD_600_) of WL03 and WL04 were measured after treatment with various concentrations of L-arabinose (Ara) and glucose (Glu). (**C**): The GFP fluorescence/OD_600_ of strain WL05 under varying concentrations of fructose (Fru), Glu, and Ara. (**D**): The PN titer was determined by sequentially subculturing from the 1st to the 100th generations. (**E**): Growth of cells in the 1st, 30th, 60th, and 100th generations. (**F**): The PN titer and cell growth (OD_600_) of strains WL04, WL08, and WL09. The data are means ± SD of three independent biological replicates (*n* = 3).

**Figure 5 microorganisms-12-00933-f005:**
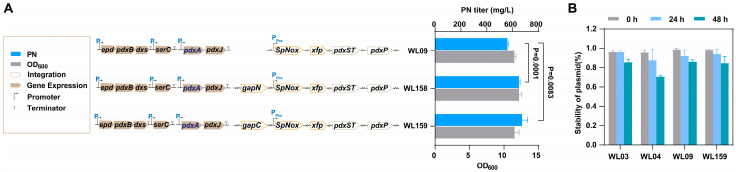
The PN titer, cell growth (OD_600_) and stability of plasmid (%) of the strains. (**A**): The PN titer and cell growth (OD_600_) of strains WL09, WL158, and WL159. (**B**): The stability of plasmid (%) of strains WL03, WL04, WL09 and WL159. The data are means ± SD of three independent biological replicates (*n* = 3), and significance (*p*-value) was assessed using a two-sided Student’s *t*-test.

**Table 1 microorganisms-12-00933-t001:** The strains used in this study.

Strain	Description	Source
MG1655	F-λ-*ilvG*-, *rfb*-50, *rph*-1	Our lab
LL006	MG1655, Δ*pdxH*::*pdxST*-2 (Bsu), Δ*pta*::Ptac-*pdxP* (Eme)	Our lab [11]
WL01	LL006, Δ*ldhA*::*xfp* (Blo)	This study
WL02	LL006 harboring p15ASI-Ptac-*epd* (Gni)-*pdxB* (Eco)-*dxs* (Eme)-P_J231119_-*serC* (Eco), pRSFDuet-1_P3-*pdxA2*-*pdxJ1*	This study
WL03	WL01 harboring p15ASI-Ptac-*epd* (Gni)-*pdxB* (Eco)-*dxs* (Eme)-P_J231119_-*serC* (Eco), pRSFDuet-1_P3-*pdxA2*-*pdxJ1*	This study
WL04	WL01 harboring p15ASI-Ptac-*epd* (Gni)-*pdxB* (Eco)-*dxs* (Eme)-P_J231119_-*serC* (Eco), pRSFDuet-1_P3-*pdxA3* (F140I)-*pdxJ1*, pBAD-SpNox	This study
WL05	WL03 harboring pBAD-GFP	This study
WL06	WL01, Δ*ykgA*::J23118-SpNox	This study
WL07	WL01, Δ*ykgA*::Pro-SpNox	This study
WL08	WL06 harboring p15ASI-Ptac-*epd* (Gni)-*pdxB* (Eco)-*dxs* (Eme)-P_J231119_-*serC* (Eco), pRSFDuet-1_P3-*pdxA3* (F140I)-*pdxJ1*	This study
WL09	WL07 harboring p15ASI-Ptac-*epd* (Gni)-*pdxB* (Eco)-*dxs* (Eme)-P_J231119_-*serC* (Eco), pRSFDuet-1_P3-*pdxA3* (F140I)-*pdxJ1*	This study
WL10-WL155	PdxA mutants (details in Appendix A)	This study
WL156	WL07, Δ*pflB*::*gapN*	This study
WL157	WL07, Δ*pflB*::*gapC*	This study
WL158	WL156 harboring p15ASI-Ptac-*epd* (Gni)-*pdxB* (Eco)-*dxs* (Eme)-P_J231119_-*serC* (Eco), pRSFDuet-1_P3-*pdxA3* (F140I)-*pdxJ1*	This study
WL159	WL157 harboring p15ASI-Ptac-*epd* (Gni)-*pdxB* (Eco)-*dxs* (Eme)-P_J231119_-*serC* (Eco), pRSFDuet-1_P3-*pdxA3* (F140I)-*pdxJ1*	This study

## Data Availability

The datasets supporting the conclusion of this article are included in the article and its Appendix A.

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
