# Peer review of "Multiple Cofactor Engineering Strategies to Enhance Pyridoxine Production in Escherichia coli"

_microorganisms, 2024, doi:10.3390/microorganisms12050933_

Round 1
Reviewer 1 Report
Comments and Suggestions for Authors
The authors present a detailed manuscript on the metabolic engineering of E. coli for the production of pyridoxine, based on their previous publications.
The manuscript would really be helped with the addition of a table describing genotype and phenotype (e.g. B6 titers) of all the results presented in the manuscript.
Additionally, the manuscript would be really helped if the authors would change their x-axis in figures 1 ,3, and 4 from their strain numbers to actual genotypes.
Additional comments:
Line 98-100, the sentence is unclear and needs rephrasing
Line 104, why is 10 mM aKG added to the media? This is a high concentration and it is not explained why this is added.
Line 110-120, how were the plasmids verified by sequencing?
Line 203, for Figure 1 see comment above about labeling x-axis
Line 230-239, why was 250ml and 30ml culturing compared?
Line 240, “Unfortunately” does not really fit in a scientific text and should be edited
Line 244-257, it would be beneficial to the manuscript if computation results (delta Gs) was tabulated with actual B6 titer results.
Line 380, for Figure 3 see comment above about labeling x-axis
Line 403, 651.99 mg/L is written in too many significant numbers. Please round.
Line 425, for Figure 4 see comment above about labeling x-axis
Reviewer 2 Report
Comments and Suggestions for Authors
The manuscript "Multiple Cofactor Engineering Strategies To Enhance 2-Pyridoxine Production in Escherichia coli" reports on metabolic and enzymatic engineering of E. coli to alleviate the NAD+ deficiency (presumably by increased NADH synthesis) that occurs during pyridoxine production. The approach described is valid and multifaceted. The results are interesting and contribute to the advancement of the field. However, the manuscript needs substantial revision.
MAIN POINTS
1) The involvement of NAD+/NADH imbalance as a cause of reduced PN production is not clear or efficiently demonstrated. In this regard, the authors should
a) Properly introduce the topic in the abstract, introduction and results section by showing the pathway with chemical structures (at least for PN synthesis reactions). The manuscript should first clearly describe the problem (provide data - see point c) and then present appropriate solutions.
b) Demonstrate that the flux through the PN production pathway is significant compared to the main NADH-producing pathways: what is the % of NADH produced by PN synthesis compared to that produced by glycolysis?
c) Start the results section by quantifying the imbalance. In this respect, the results shown in Figure 3a are striking. In my opinion, they indicate problems in NAD synthesis or degradation, which are recovered in the WL04 strain. The NAD+/NADH and NAD levels should be measured for all the main strains obtained.
2) The enzyme engineering approach is interesting but difficult to understand for non-experts. Both the relative methods (sections 2.7 and 2.8) and the results should be shortened, better organised and explained, concentrating the description on the most important results. Also, the overall aim of enzyme engineering is not clearly defined; is there a competition between NAD+ and HTP that needs to be overcome? is there a need to reduce NAD+ binding? I therefore ask the authors to shorten and clarify the text.
3) The expression level of SpNOX in all cited strains should be measured directly and not via GFP. The authors should clearly demonstrate that increased expression of SpNOX results in decreased cell viability or metabolic efficiency.
4) The authors clearly show that episomal expression of the PN synthetic pathway leads to strain instability and therefore they integrate the pathway in the bacterial chromosome. However, they do not show that they increase the stability of the strain in this way. They report an increase in titer (Figure 3F) which does not appear to be significant (no statistics are provided). Maybe an increase in strain stability would add significance to their results.
5) The authors report that the introduction of GapN and GapC reduces the formation of by-products. However, they do not report any measurements or information on process by-products and yields.
LESSER POINTS
- Section 2.3 should be better explained (e.g. "seamless cloning"). No reference is given for some of the methods reported.
- Paragraph 2.4. The buffer used and the approach in general should be better explained. Do the authors use different buffers to differentially degrade NAD+ or NADH? This is not clear from the text.
- Figure 1B would be much clearer if the genetic modifications were given in addition to the strain names.
- Please better describe strain LL006
- Section 3.2. The increase in production is not clear (the authors should first state that they are using the titre to assess the increase in production; elsewhere they introduce "productivity"). They first state that there is an increase of 40% (line 230, then 10% in line 10, then 26% in line 241). Are they reporting different parameters? If so what each of them means?
- Line 334; what is the reference to "7.5%"?
Comments on the Quality of English Language
TYPOS - SENENTECS TO BE REFRASED
In general, the English used needs to be revised. The following words/sentences need to be rephrased.
- line 29 "... by themeselves..."
- line 48 "...strain degradation..."
- line 56 "By substituting...."
- line 227 do "..lowest" indicate "most negative"?
- rephrase the title of par. 3.2
- line 281; what is the "...duration of the protein..."?
- Figure 2 please define RMSD and MD in the legend
- line 312-313 "NADH oxidase......"
- line 427 ".... availability of ...."
- line 442 "...we are not yet clear...."
Round 2
Reviewer 2 Report
Comments and Suggestions for Authors
I thank the authors for their efforts, but some of their responses are not complete.
Importantly, they have not properly addressed my main concern:
Is there a NADH imbalance or a NAD (NAD+ + NADH) concentration problem (I think data mostly show this)? If so, how can this be explained? Furthermore, I think it is possible to roughly estimate glycolytic NADH vs. PN synthesis NADH by comparing the rate of glucose uptake with the rate of PN production. I think this would allow the authors to assess whether the NADH formed during PN synthesis is significant compared to glycolytic NADH.
- I do not understand what the authors mean by "metabolic driving force". Since the authors are trying to optimise an enzyme, the problem should be kinetic and not that of increasing the "driving force". So I think "increasing the catalytic efficiency of the rate-limiting enzyme pdxA" is a better definition. Furthermore, it is still not clear why reducing NAD+ binding (if this is the aim of the engineering process) can increase catalytic efficiency. This should be better defined in the text. In addition, they report that "... the mutant design of the PdxA enzyme improved the combination of 4HTP..."; what do the authors mean? perhaps the binding of 4HTP? In summary I ask to clearly and concisely define the aim of the engineneering process and approach used.
- I still do not understand all the efforts the authors made to regulate the expression of SpNox; is the expression of this enzyme toxic? can the authors provide data on this? alternatively is everything based on PN production titres? are those significantly different? this point should be discussed more clearly.
- The authors reply that ".... increased the stability of the strain to some extent" by integrating SpNox into the chromsome; they should provide data.
- The authors report that "..., α-KG can be converted back to glutamine in an NADPH-dependent reaction..."; glutamine or glutamate? Which reaction? Please be more specific.
- I appreciate the authors' efforts to provide more details in the Methods section. However, the added text should be revised. These are just example of sentence which need to be rephrased: lines 129-131; l.147 "..absorbing the supernatant"; l.153 "...sample was inhaled ..". l.332 "..Compared with the group adding 0.2 mM...". I emphasise that these are only examples and that the revision must be thorough.
Comments on the Quality of English LanguageAn extensive revision is required expecially, but not exclusively, for the newly added parts.
Author Response
请参阅附件。
